# Neglected Comorbidity of Chronic Heart Failure: Iron Deficiency

**DOI:** 10.3390/nu14153214

**Published:** 2022-08-05

**Authors:** Hana Manceau, Jérome Ausseil, Damien Masson, Jean-Paul Feugeas, Bernard Sablonniere, Régis Guieu, Hervé Puy, Katell Peoc’h

**Affiliations:** 1Inflammation Research Center CRI, INSERM UMRs 1149, Université de Paris, 75018 Paris, France; 2Departement of Biochemistry, Bichat Hospital, APHP, 75018 Paris, France; 3Laboratory of Biochemistry and Molecular Biology, CHU Toulouse, Infinity, INSERM UMR 1291, CNRS UMR 5051, Université Paul Sabatier, CEDEX 3, 31024 Toulouse, France; 4Department of Biochemistry, Faculté de Médecine, Nantes University Hospital, 44000 Nantes, France; 5Laboratoire de Biochimie Hôpital Jean Minjoz, Université Bourgogne Franche-Comté, INSERM, EFS BFC, UMR1098, Interactions Hôte-Greffon-Tumeur/Ingénierie Cellulaire et Génique, F-25000 Besançon, France; 6Centre de Biologie Pathologie et Génétique, CHU Lille, JPArc—Centre de Recherche Jean-Pierre AUBERT, Institut de Biochimie et Biologie moléculaire, University Lille, Inserm, CHU Lille, UMR-S 1172, F-59000 Lille, France; 7Laboratory of Biochemistry, Assistance Publique des Hopitaux, Center for Cardio-Vascular and Nutrition Research, INSERM, INRAE, Aix-Marseille University, 13005 Marseille, France

**Keywords:** iron deficiency, heart failure, transferrin saturation coefficient, serum ferritin, ferric carboxymaltose

## Abstract

Iron deficiency is a significant comorbidity of heart failure (HF), defined as the inability of the myocardium to provide sufficient blood flow. However, iron deficiency remains insufficiently detected. Iron-deficiency anemia, defined as a decrease in hemoglobin caused by iron deficiency, is a late consequence of iron deficiency, and the symptoms of iron deficiency, which are not specific, are often confused with those of HF or comorbidities. HF patients with iron deficiency are often rehospitalized and present reduced survival. The correction of iron deficiency in HF patients is associated with improved functional capacity, quality of life, and rehospitalization rates. Because of the inflammation associated with chronic HF, which complicates the picture of nutritional deficiency, only the parenteral route can bypass the tissue sequestration of iron and the inhibition of intestinal iron absorption. Given the negative impact of iron deficiency on HF progression, the frequency and financial implications of rehospitalizations due to decompensation episodes, and the efficacy of this supplementation, screening for this frequent comorbidity should be part of routine testing in all HF patients. Indeed, recent European guidelines recommend screening for iron deficiency (serum ferritin and transferrin saturation coefficient) in all patients with suspected HF, regular iron parameters assessment in all patients with HF, and intravenous iron supplementation in symptomatic patients with proven deficiency. We thus aim to summarize all currently available data regarding this common and easily improvable comorbidity.

## 1. Introduction

Heart failure (HF) is the inability of the myocardium to provide sufficient blood flow to meet the body’s metabolic needs. The main clinical signs of HF are shortness of breath, severe fatigue, and lower-extremity edema. According to the 2016 European Society of Cardiology recommendations, three categories of HF are defined according to the left ventricular ejection fraction (LVEF) value: reduced ejection fraction (LVEF < 40%), preserved (LVEF > 50%), and intermediate (LVEF of 40–50%) HF. The severity of HF is often semi-quantified with the New York Heart Association (NYHA) functional stages, from stage I (no limitation) to stage IV (extreme restriction).

HF can be the consequence of ischemic (infarcts, angina), toxic (alcohol, chemotherapy), or infectious myocardial pathologies; it can also result in abnormal loading conditions (hypertension, valvulopathy) or infiltrative pathology (amyloidosis, sarcoidosis, hemochromatosis) [1].

HF is the most common cardiovascular disease, with an estimated prevalence of 1–2% of the adult population in developed countries (10% in the age group >70 years) [1]. The incidence of chronic HF increases with the aging of the population and the improved management of other cardiovascular diseases. However, diagnosis can be difficult. Comorbidities are frequent, and symptoms can be non-specific in early HF. The improvement of diagnostic methods leads to an earlier detection of the disease and could then prevent its progression. Thus, natriuretic-peptide measurements are essential for diagnosing and monitoring HF. Natriuretic peptides (BNPs) or the inactive fragment of pro-BNPs (NT pro-BNPs) are solely produced by the cardiac tissue, and their concentrations increase with the cardiac-wall stress in proportion to the severity of HF [1]. Although they are correlated, their optimal cut-off concentrations differ and may vary with age (see [2] for complete recommendations).

The evolution of chronic HF is punctuated by hospitalizations related to acute episodes of cardiac decompensation. Each of these episodes worsens HF; among hospitalized patients, the 5-year mortality rate is high (75%), as is the rate of rehospitalization (about 80%) [3]. In its July 2021 report “Improving the Quality of the Healthcare System and Controlling Expenditures,” the French “Caisse Nationale d’Assurance Maladie” proposed a 15% reduction in HF-related rehospitalizations by 2022 [4]. Indeed, more than 1.5 million patients (16% aged 85 and over) are affected by this pathology, with 165,000 hospitalizations each year and a cost of approximately EUR 3 billion. The reduction in the rate of rehospitalization after an episode of cardiac decompensation is a new national indicator included in “Contract for the Improvement of the Quality and Efficiency of Care” (CAQES) [5]. Therefore, all methods or therapies that contribute to reducing the hospitalization time of patients with HF should be considered. Iron deficiency, often associated with HF [6], can be easily detected, and its treatment is simple and effective. Indeed, the randomized clinical trials described below have shown that iron supplementation reduces the hospitalization rate of HF patients.

## 2. Iron Deficiency, the Most Common Comorbidity of Heart Failure

HF is frequently associated with comorbidities, the most common being chronic kidney disease (40%) [7], diabetes (30–40%) [8], and chronic obstructive pulmonary disease (20.5%) [9]. These comorbidities have a significant impact on hospitalizations and mortality.

Iron deficiency is often associated with chronic HF [10,11], with or without anemia, as the most common nutritional deficiency. Indeed, iron deficiency is most often discovered in anemia [12]. Iron-deficiency frequencies from 37% to 61% were reported in different studies [13,14,15]. In the study by Klip et al. on European cohorts of 1500 HF patients, iron deficiency was diagnosed in 61.2% of patients with anemia and 45.6% without anemia [16]. The CARENFER study recently conducted in France on 1661 HF hospitalized patients reported that 49.6% had iron deficiency [17].

Even in the absence of anemia, iron deficiency is a poor prognostic factor in HF [18]. Iron deficiency increases the relative risk of death by 40–60%. For example, a prospective study on nearly 550 patients with NYHA class II–III chronic HF (mean LVEF of 26%) reported that the adjusted relative risk of the composite endpoint of all-cause death or heart transplantation was increased by 58% hen iron deficiency was present. In contrast, anemia was not an independent risk factor [19]. Another study in the United Kingdom included 150 patients with HF [20]. Compared with patients without anemia and iron deficiency, the relative risk of death was not significantly increased in anemic patients without iron deficiency.

In contrast, it was twice as high in nonanemic patients with iron deficiency [21]. In the European-cohort analysis by Klip et al., iron deficiency was an independent mortality risk factor (relative risk increased by 42% in multivariate analyses), along with the classic risk factors (sex, age, NYHA class, diabetes, hypertension, etc.) [16]. Anemia was not an independent risk factor.

## 3. Iron Is an Essential Element for the Correct Functioning of the Heart Muscle

Iron plays a significant role in erythropoiesis and oxygen transport. However, it also participates in DNA replication and repair, cell growth and differentiation, brain function, dioxygen storage in myoglobin, and energy metabolism of striated muscles and heart (ATP synthesis) [22,23] (Figure 1). Cardiomyocytes are characterized by a high myoglobin concentration in the cytosol and contain numerous mitochondria, which produce the energy necessary for cardiac-muscle contractions. Iron plays a significant role in mitochondrial functions as a cofactor in iron–sulfur-cluster-containing proteins, heme-containing proteins, and iron-ion-containing proteins (Figure 2). Approximately 90% of the ATP required for the proper functioning of the heart muscle (i.e., for contraction) is produced by the mitochondrial enzymatic complexes of the respiratory chain [24]. Cellular iron deficiency was shown to result in reduced activity of Fe-S-cluster-based complexes in the mitochondria of human cardiomyocytes and to be associated with impaired mitochondrial respiration and morphology, ATP production, and contractility (Figure 1) [25,26]. Interestingly, restoring intracellular iron concentrations can reverse these effects on muscle [26,27].

This essential role of iron as a cofactor in the structure of proteins involved in oxidative phosphorylation and anti-oxidative enzymes (Figure 2) impacts the pathophysiology of progressive cardiac remodeling in HF patients [19,21,28]. Martens et al. included patients with HF and iron deficiency receiving cardiac-resynchronization therapy [29] and evidenced that iron supplementation improved cardiac function and the cardiac-force–frequency relationship [19]. Intravenous (IV) iron supplementation improved cardiac remodeling, particularly via a significant increase in LVEF. The heart muscle is consequently susceptible to iron deficiency, but those deleterious effects can be corrected—at least in part—by iron supplementation [30].

## 4. Iron, a Finely Regulated Element during Inflammation: Role of Hepcidin

Most of the iron in the body of an adult (about 4000 mg) is contained in the hemoglobin of red blood cells (1800 mg) and their precursors (300 mg), and 10–15% is included in myoglobin and various enzymes [31]. Iron is also stored in liver parenchymal cells (1000 mg); macrophages of the reticuloendothelial system (liver, spleen, bone marrow) temporarily store iron recycled from senescent red blood cells (600 mg).

Intestinal iron absorption occurs at the apical pole of enterocytes in the duodenum and proximal jejunum. Iron is transported to the basal pole of the enterocyte and released into the plasma by ferroportin [32]. This polarized iron transport (1–2 mg/day) is saturable and limited to 10 mg/day. In the circulation, iron is taken up by transferrin and distributed to target organs whose cells express the transferrin receptor (RTf or CD71) [33,34]. The transferrin/RTf complex is internalized through endocytosis, and iron is released into the cell. Iron can participate in the metabolism or be stored as ferritin in the cytosol.

Hepcidin is a liver-produced peptide, particularly produced during infectious and inflammatory episodes. This molecule is the primary systemic regulator of iron homeostasis. Hepcidin binds to the transmembrane ferroportin, inducing ferrorportin degradation and decreasing intracellular iron export. Hepcidin expression increases with the serum iron concentration [14]. When a large amount of iron is orally absorbed (i.e., >40 mg), the serum hepcidin concentration increases [35]. The export of intracellular iron from macrophages, hepatocytes, and enterocytes is blocked; iron remains sequestered in the cell, and intestinal absorption is inhibited [36].

In chronic inflammatory diseases, the increased synthesis of hepcidin produced in response to inflammatory cytokines (IL-6 in particular) blocks the absorption of oral iron of enterocytes, thus rendering oral supplementation ineffective. Only the parenteral route allows the blockage of enterocyte absorption to be bypassed. The parenteral route makes it possible to overcome the sequestration of iron. This iron is stored in hepatocytes and macrophages of the reticuloendothelial system that participate in the recycling of red blood cells [37]. In addition, iron administered via the IV route increases the expression of ferroportin on macrophages [38], thus promoting the export of iron from the storage compartment to circulation. Thus, the choice of the iron administration route is notably dictated by inflammation.

## 5. Biological Diagnosis of Iron Deficiency

Understanding the pathophysiological mechanisms of iron deficiency is essential for the rational use of diagnostic biomarkers and the therapeutic approach. Two forms of iron deficiency must be distinguished: absolute iron deficiency is the consequence of a quantitative decrease in iron stores, whereas functional iron deficiency is due to iron sequestration in otherwise quantitatively regular or abundant stores. The clinical implications of these two forms of iron deficiency are identical.

Iron deficiency in HF is frequently both absolute and functional [30]. Absolute iron deficiency may result from appetite loss, decreased absorption due to intestinal edema, or long-term hemorrhages related to antiplatelet agents, nonsteroidal anti-inflammatory drugs, or anticoagulants. HF is also characterized by systemic inflammation leading to functional iron deficiency.

In daily practice, the most valuable biomarkers for assessing iron status are serum ferritin, a marker of iron stores, and transferrin saturation (TSAT), which indicates the amount of iron transported in the circulation and available for cellular metabolism. TSAT is calculated from serum iron and transferrin; a low TSAT value is an essential diagnostic criterion for iron deficiency, whether absolute or functional. Thus, a recent retrospective study included 1701 patients hospitalized for decompensated HF, and an iron workup was performed within 24–72 h after hospitalization. The risk of 30-day readmission for HF or death (composite criterion) was higher the lower TSAT was [39].

The reliability of serum-ferritin measurement may be compromised in patients with chronic diseases or other conditions associated with inflammation. Therefore, there is a broad consensus on the recommendations to assess TSAT with serum ferritin [40]. According to the French National Authority for Health recommendations, only serum ferritin should be performed as a first-line test for iron deficiency; in inflammatory conditions, chronic renal disease, or malignant diseases, serum ferritin and TSAT are indicated [41]. There is no indication to measure iron concentration alone.

There is no consensus on the thresholds. Still, recent international guideline recommendations have shown agreement in chronic diseases to define iron deficiency with serum ferritin < 100 µg/L and/or TSAT < 20% [12,36]. Thus, according to the European Society of Cardiology recommendations, a serum ferritin concentration < 100 µg/L defines absolute iron deficiency in HF, and serum ferritin of 100–299 µg/L plus TSAT < 20% represents functional iron deficiency [42,43].

Iron deficiency is associated with the overexpression of the transferrin receptor on the cell surface and, consequently, with an increase in the serum concentration of the soluble transferrin receptor (RsTf) [44]. The measurement of the serum RsTf concentration is limited to rare situations.

Iron deficiency is often missed because the causative disease or comorbidities may mask it. Indeed, before iron-deficiency anemia, iron deficiency manifests itself via unspecific symptoms, the most frequent of which is fatigue. Therefore, iron deficiency remains underscreened despite its high frequency in HF (nearly 50% of patients) and poor prognosis [43]. In a 2017 survey in France on 2822 patients hospitalized for HF at least once in the previous five years, only 38.1% had had a diagnostic test [45]. In another study including 1484 patients with HF from Germany and Switzerland, the iron status was determined in 62.2% despite the centers’ participation in a registry devoted to iron deficiency [46].

In the Swedish HF registry, of 21,496 patients with HF, only 27% had had a diagnostic test [47]. Another study from a French medico-administrative database from 2006 to 2013 of iron-supplemented patients showed that TSAT was not assessed and serum ferritin only rarely so, including patients with chronic inflammation [48].

## 6. Iron Supplementation in Heart Failure

In case of iron deficiency, iron can be provided either via the oral or parenteral route. Although cheap and readily available, the oral route is frequently (up to 70%) associated with poor gastrointestinal tolerance and various levels of efficacy and compliance [49,50]. The IV route of new-generation iron preparations [51] might be associated with limited gastrointestinal side effects, injection-site reactions, and infrequent anaphylactic reactions; the safety is considered quite satisfactory [52].

Published in 2017, the double-blind IRONOUT-HF study on 225 patients with chronic stable HF (NYHA classes II–IV) and iron deficiency showed that oral iron administration (150 mg × 2/day) for 16 weeks was comparable to placebo in terms of functional capacity as assessed via peak oxygen consumption and walking [36]. In contrast, the FAIR-HF, CONFIRM-HF, FERRIC-HF, and EFFECT-HF studies demonstrated the efficacy of IV iron on HF symptoms [53,54,55,56]. This difference in the effectiveness of the oral and IV routes is explained by a high concentration of hepcidin in chronic HF [36].

In the AFFIRM-HF study, the effect of IV iron supplementation on mortality was evaluated in chronic HF patients (LVEF < 50%) with iron deficiency (according to the 2016 ESC criteria) hospitalized for an acute episode of HF [54]. At discharge, stabilized patients were randomized to IV iron carboxymaltose or placebo. After a 52-week follow-up (1108 evaluable patients), the relative risk of hospitalization for a new decompensation was reduced by 26%, without significant effects on cardiovascular mortality.

Jankowska et al.’s [20] meta-analysis included five randomized studies totaling 851 patients with systolic HF and iron deficiency. Patients were treated with IV iron or a comparator (oral or IV placebo, oral iron). In all patients (with or without anemia), the relative risk of composite endpoint “all-cause death or hospitalization for cardiovascular reasons” was reduced by 56%, and the relative risk of composite endpoint “cardiovascular death or hospitalization for progression of chronic heart disease” was reduced by 61%.

Another meta-analysis by Anker et al. [57] included four randomized [54,57,58,59], double-blind studies (FER-CARS-01, FAIR-HF, EFFICACY-HF, and CONFIRM-HF) with a total of 839 patients with chronic systolic HF and iron deficiency to compare IV iron carboxymaltose and placebo. The relative risk of cardiovascular hospitalization or death was reduced by 41% in the iron-supplemented group.

IV iron supplementation, therefore, has a demonstrated clinical benefit in iron-deficient chronic HF patients, anemic or not. Iron deficiency should be considered an independent therapeutic goal in this population [13].

According to 2017 US recommendations, IV iron supplementation for iron deficiency has to be considered in NYHA II–III patients [60]. The iron status should be reassessed during routine visits (once or twice a year) and after each hospitalization for HF. The recently updated European Society of Cardiology guidelines (2021) indicate that IV iron carboxymaltose therapy should be considered in patients with symptomatic chronic HF with LVEF ≤ 45% and iron deficiency defined as serum ferritin < 100 µg/L or serum ferritin at 100–299 µg/L with TSAT < 20% to alleviate symptoms, and improve function and quality of life [61]. Iron supplementation with IV iron carboxymaltose should also be considered in patients recently hospitalized for HF with LVEF < 50% and iron deficiency according to the exact definition to reduce the risk of HF-related rehospitalization

## 7. Conclusions

Iron deficiency is the most common comorbidity of HF, and its negative impact is well documented. The efficacy of IV iron supplementation on disease progression and rehospitalizations has also been demonstrated. The diagnosis and treatment of iron deficiency are now well defined, and the most recent European recommendations consider iron deficiency in HF. In particular, in patients hospitalized for cardiac decompensation, the 2021 European recommendations call for correcting possible iron deficiency with IV iron at hospital discharge. Therefore, serum ferritin plus TSAT should now be part of routine testing in all HF patients.

## Figures and Tables

**Figure 1 nutrients-14-03214-f001:**
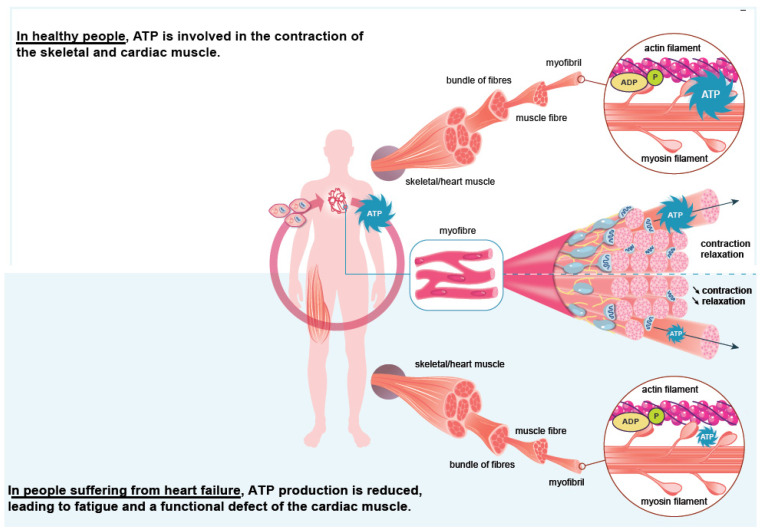
Role of iron in the heart muscle’s functioning. Iron is notably mandatory for the energy production involved in heart contractions and for the oxygenation of the muscle, which ensures its correct function over time.

**Figure 2 nutrients-14-03214-f002:**
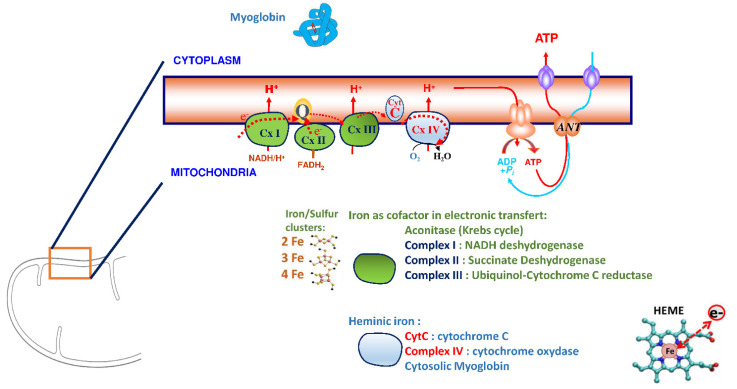
Role of the iron outside of hematopoiesis. Iron is an essential cofactor in most proteins involved in oxidative phosphorylations and anti-oxidative enzymes. As shown in the present figure, iron is a constitutive element of numerous proteins, either as a component of the heme ring in hemoproteins (such as Myoglobin, involved in muscle oxygenation, or cytochromes, involved in oxido-reductive reactions) or in iron–sulfur clusters involved in electronic transfers.

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
