# Peer review of "Neglected Comorbidity of Chronic Heart Failure: Iron Deficiency"

_nutrients, 2022, doi:10.3390/nu14153214_

Round 1

Reviewer 1 Report

Hana Manceau and coworkers wrote a useful review giving the reader an overview of iron deficiency as the most common comorbidity of heart falure. The review shows all aspects of the iron metabolism and deficiency in heart insufficiency, describing also the difference between functional and absolute iron deficiency. All the important resources were referenced to databases. The better knowledge of the pathophysiological mechanisms of iron deficiency is vital for the rational use of diagnostic biomarkers and the therapeutic approach in heart failure.

There are few comments which may be useful:

Page 4

Line 133 and 134: Sentence ‘Martens et al. included patients with HF and iron deficiency receiving cardiac resynchronization therapy.’ is not clear (it seems to be somehow cut off), please change it.

Line 134: Does IV mean intravenous ? Abbreviation IV might be not clear for every reader.

Line 153 and 154: Ferroportin is responsible for exporting iron from the cell, so this fragment ‘inhibition of intracellular iron export’ should be changed.

Line 157: Sentence ‘iron is maintained in the supply’ is not clear. Please change it.

Page 5

Line 162 to 165: The information described in this place can not be found in the cited reference No 31. Please correct it.

Line 165 to 167: The relevant reference is missing, please add it.

Page 6

Line 257 to 264: Is it really necessary to show Table 1, which can be found in identical version in Reference No 31? What is more, the references which can be found in this table in original publication, here are  disappearing.  I would recomend to remove this table.

Author Response

Thanks a lot for this deep reading and all the advices you give us to improve the manuscript.

Here is a point-by-point response.

There are few comments which may be useful:

Page 4

Line 133 and 134: Sentence ‘Martens et al. included patients with HF and iron deficiency receiving cardiac resynchronization therapy.’ is not clear (it seems to be somehow cut off), please change it.

We have modified the text as follows

Martens et al. included patients with HF and iron deficiency receiving cardiac resynchronization therapy [24] and evidenced that iron supplementation improves cardiac function and cardiac force-frequency relationship [17].

Line 134: Does IV mean intravenous ? Abbreviation IV might be not clear for every reader.

We have introduced the Intravenous (IV) abbreviation.

Line 153 and 154: Ferroportin is responsible for exporting iron from the cell, so this fragment ‘inhibition of intracellular iron export’ should be changed.

The sentence has been changed as follows:

Hepcidin binds to the transmembrane ferroportin, inducing the ferroportin degradation and decreasing intracellular iron export.

Line 157: Sentence ‘iron is maintained in the supply’ is not clear. Please change it.

The sentence has been changed as follows

Iron remains intracellular

Page 5

Line 162 to 165: The information described in this place can not be found in the cited reference No 31. Please correct it.

The sentence has been changed as follows : 

This iron is stored in hepatocytes and macrophages of the reticuloendothelial system that participate in recycling red blood cells [32].

Line 165 to 167: The relevant reference is missing, please add it.

We introduced a new reference: In addition, iron administered by the IV route increases the expression of ferroportin on macrophages [33].

Page 6

Line 257 to 264: Is it really necessary to show Table 1, which can be found in identical version in Reference No 31? What is more, the references which can be found in this table in original publication, here are  disappearing.  I would recomend to remove this table.

We have removed the table, as recommended.

Reviewer 2 Report

This article is well organized and presented in a clear and complete way. The figures are clear. The table shown summarizes the recommendations proposed by the European Society of Cardiology 2021. The bibliography shown is relevant to the text.

Author Response

Thank you for this nice comment. We are grateful for your help.

Reviewer 3 Report

Dear authors,

The manuscript “Neglected comorbidity of chronic heart failure: iron deficiency” presented an interesting review on iron deficiency in heart failure.

Please find my suggestions and comments below:

·        Lines 62-64: authors should present some aspects regarding BNP (secretion, role, cut off value in HF, etc.)

·        Figure 2: authors should provide more explanations in figure caption

·        RsTf explanation should be provided in line 203, not 204

·        Lines 210-213: other worldwide studies on diagnostic tests for iron deficiency should be added.

·        Lines 216-218: adverse reactions of parenteral iron should be presented

·        Line 51, Lines 165-167, and Lines 216-218: some references should be added

Author Response

Thanks a lot for your time and the help you provide us to improve our manuscript.

Here is a point-by-point reponse: 

The manuscript “Neglected comorbidity of chronic heart failure: iron deficiency” presented an interesting review on iron deficiency in heart failure.

Please find my suggestions and comments below:

  • Lines 62-64: authors should present some aspects regarding BNP (secretion, role, cut off value in HF, etc.)

We have modified the text as follows

Natriuretic peptide (BNP) or the inactive fragment of pro-BNP (NT pro-BNP) are solely produced by the cardiac tissue, and their concentrations increase with the cardiac wall stress in proportion to the severity of HF [1]. Although they are correlated, their optimal cut-off concentrations differ and may vary with age (see [2] for complete recommendations).

  • Figure 2: authors should provide more explanations in figure caption

We have developed this part as follows:

Figure 2. Role of the iron outside of hematopoiesis: iron is an essential cofactor in proteins of oxidative phosphorylation and anti-oxidative enzymes. As shown in the present figure, iron is a constitutive element of numerous proteins, either as a component of the heme ring in the hemoproteins (such as Myoglobin, implied in the muscle oxygenation, or cytochromes implied in oxido-reductive reactions) or in Iron-Sulfur clusters implied in electronic transfers. Iron is thus directly implied in muscle oxygenation independently from hemoglobin through myoglobin as a cofactor in proteins of oxidative phosphorylation and anti-oxidative enzymes.

  • RsTf explanation should be provided in line 203, not 204

We have corrected this point

Iron deficiency is associated with overexpression of the transferrin receptor on the cell surface and, consequently, an increase in the serum concentration of the soluble transferrin receptor (RsTf) [36]. The measurement of the serum RsTf concentration is limited to rare situations.

  • Lines 210-213: other worldwide studies on diagnostic tests for iron deficiency should be added.

We have followed this recommendation :

In another study including 1,484 patients with HF from Germany and Switzerland, iron status was determined in 62.2% despite the centers' participation in a registry devoted to iron deficiency [40].

In the Swedish HF registry, of 21 496 patients with HF, only 27% had a diagnostic test [41].

  • Lines 216-218: adverse reactions of parenteral iron should be presented

We have corrected this point:

The IV route of new generations’ iron preparations [43] might be associated with limited gastrointestinal side effects, injection site reactions, and infrequent anaphylactic reactions; the safety is considered quite satisfactory [44].

  • Line 51, Lines 165-167, and Lines 216-218: some references should be added

We have introduced a reference in each part :

In addition, iron administered by the IV route increases the expression of ferroportin on macrophages [33]. The oral route, although cheap and readily available, is frequently (up to 70%) associated with poor gastrointestinal tolerance, together with various efficacy and compliance [41,42].